# Structural Phase Transitions and Thermal Degradation Process of MAPbCl_3_ Single Crystals Studied by Raman and Brillouin Scattering

**DOI:** 10.3390/ma15228151

**Published:** 2022-11-17

**Authors:** Furqanul Hassan Naqvi, Jae-Hyeon Ko

**Affiliations:** School of Nano Convergence Technology, Nano Convergence Technology Center, Hallym University, Chuncheon 24252, Republic of Korea

**Keywords:** lead halide perovskites, MAPbCl_3_, Raman spectroscopy, phase transitions

## Abstract

Raman spectroscopy was applied to MAPbCl_3_ single crystals in a wide frequency range from 10 to 3500 cm^−1^ over a broad temperature range from −196 °C to 200 °C including both two structural phase transitions and a thermal degradation range. Low-frequency lattice modes of MAPbCl_3_ were revealed for the first time, which showed discontinuous anomalies along with the change in the number of Raman modes at the transition points of −114 °C and −110 °C. Several Raman modes related to the C–N stretching and MA rocking modes in addition to the lattice modes displayed temperature dependences similar to those of MAPbBr_3_ in both Raman shifts and half widths, indicating that the MA cation arrangement and H–halide bond interactions behave similarly in both systems during the phase transition. The substantial increase in the half widths of nearly all Raman modes especially suggests that the dynamic disorder caused by the free rotational motions of MA cations induces significant anharmonicity in the lattice and thus, reduces the phonon lifetimes. High-temperature Raman and Brillouin scattering measurements showed that the spectral features changed drastically at ~200 °C where the thermal decomposition of MAPbCl_3_ into PbCl_2_ began. This result exhibits that combined Raman and Brillouin spectroscopic techniques can be a useful tool in monitoring temperature-induced or temporal changes in lead-based halide perovskite materials.

## 1. Introduction

Lead-based halide perovskites (LHPs) with an ABX_3_ stoichiometry consist of CH_3_NH_3_ (methylammonium, MA) or CH(NH_2_)_2_ (formamidinium, FA) cations at the A site, Pb at the B site, and Cl, Br, or I at the X site. The high compositional flexibility of these perovskites allows us to have various analogs and large degrees of freedom in tuning the physical properties. The most widely investigated organic–inorganic composition of LHPs is MAPbX_3_ (X = Cl, Br, I) where MA cations are centered inside the lead halide cage, whereas inorganic cations, such as Cs, can also occupy the same sites.

LHPs have shown great functional significance in photovoltaics and optoelectronics [1]. Furthermore, they have attracted widespread attention due to their fascinating properties such as long carrier diffusion length [2], low trap density [2], large absorption coefficients [3], low-cost fabrication routes, and solution processability [4]. Due to these merits, there has been a rapid increase in the use of LHPs in many applications such as solar cells [5], photodetectors [6], and light-emitting diodes [7]. For instance, recent studies on perovskite solar cells have demonstrated an extraordinary optoelectronic performance showing a dramatic increase in the photoconversion efficiency up to ~25% in a few years [8]. Furthermore, LHPs are tempting technological materials to be used as photocatalysts [9] and as a probe in bioimaging field [10]. Particularly, methylammonium lead chloride (MAPbCl_3_), due to its wide band gap, is famous for ultraviolet photodetection [11].

The structural changes and phase transitions in hybrid perovskites account for the change in macroscopic properties which are associated with device performances [12]; for example, dielectric permittivity changes due to symmetry breaking induced by the distortion of the inorganic octahedral structure and the displacement of organic cations [13]. Owing to these facts, the study of structural phase transitions is of great significance. In order to investigate the structural changes in these perovskites, various techniques such as PL (photoluminescence) spectroscopy [14], XRD (X-ray diffraction) [15], Raman spectroscopy [16], NMR (nuclear magnetic resonance) spectroscopy [17], and Brillouin spectroscopy [18] have previously been employed.

MAPbCl_3_ shows a high bandgap energy of 2.88 eV and is, thus, transparent, which is ideal for visible light spectroscopy [19]. MAPbCl_3_ undergoes two major phase transitions upon decreasing temperature from cubic to tetragonal and then from tetragonal to orthorhombic phase [16]. Many previous reports have studied the phase transition temperatures and the optical phonon modes of MAPbCl_3_ through Raman spectroscopy [15,16,20]. We summarized in Table 1 some of the major previous research works that studied the phase transition temperatures by employing different techniques such as Raman, calorimetric, IR (infrared), XRD, and terahertz spectroscopy [15,16,21,22,23,24,25]. It shows that some discrepancies still remain among the reported phase transition temperatures. In addition, previous Raman studies lack a complete analysis of the optical phonon modes in a wide frequency range as a function of temperature. We thus sought to investigate the structural phase transitions of this material in more detail through a complete Raman spectroscopic investigation. Maleej et al. conducted one of the first Raman spectroscopic investigations on MAPbCl_3_ [20], where the authors reported the evolution of Raman spectra but in a limited frequency range and at only at a few temperatures. Recently, Nguyen et al. discussed the temperature-dependent changes in the Raman spectra of MAPbCl_3_ but lacked a discussion of the low-frequency modes [17].

The aim of our work was to extend the Raman investigation to all optical phonons of MAPbCl_3_ in a full frequency range of 10–3500 cm^−1^ and a wide temperature range from −190 °C to 20 °C. To our knowledge, this is the first study involving the detailed discussion of temperature-dependent Raman modes of MAPbCl_3_ in the widest frequency range. The optical phonons probed by Raman spectroscopy are sensitive to structural changes and, thus, detailed mode analysis may clarify the phase transition behaviors of this interesting material. Furthermore, besides all the beneficial properties, the extensive commercial use of LHPs is hindered by their fast degradation and low chemical and mechanical stability [26,27]. These perovskites are unstable under harsh environmental conditions such as high temperatures and humidity, which cause degradation in device performances. It is thus very important to monitor the degradation process of MAPbCl_3_ by various methods. In this study, we performed, for the first time, both high-temperature Raman and Brillouin scattering measurements to monitor the thermal stability of MAPbCl_3_ single crystals.

## 2. Experimental Section

### 2.1. Precursors

Lead chloride (PbCl_2_, 99.999%), hydrochloric acid (HCl, 37%, ACS reagent), diethyl ether (HPLC grade, ≥99.9%), dimethyl sulfoxide (DMSO, anhydrous ≥99.9%), methylamine (CH_3_NH_2_, 40% in water), and ethanol (anhydrous 99.5%) were purchased from Merck Korea Ltd. (Seoul, Republic of Korea). All of these chemicals were used as received, without any further purification.

### 2.2. Single Crystal Synthesis

The synthesis of MAPbCl_3_ single crystals consisted of a two-step process. The first step involves preparing methyl ammonium (MACl, CH_3_NH_3_Cl), and the second step involves the crystallization of MAPbCl_3_. The equations below show the two-step reaction process:$${\mathrm{CH}}_{3}{\mathrm{NH}}_{2}+\;\mathrm{HCl}\;=\;{\mathrm{CH}}_{3}{\mathrm{NH}}_{3}\mathrm{Cl}$$
$${\mathrm{CH}}_{3}{\mathrm{NH}}_{3}\mathrm{Cl}\;+\;{\mathrm{PbCl}}_{2}=\;{\mathrm{CH}}_{3}{\mathrm{NH}}_{3}{\mathrm{PbCl}}_{3}$$

Figure 1 shows a schematic diagram of the complete synthesis process of MAPbCl_3_ single crystals. Firstly, to synthesize MACl, we added methylamine in a round bottom flask and placed it in an ice bath. Then, HCl (24.6 mL) was added dropwise to methylamine (30.6 mL) according to the molar ratio of CH_3_NH_2_:HCl = 1.2:1. The purpose of the ice bath was to maintain the reaction temperature. The mixture was kept under constant stirring for 2 h until the solution was completely dissolved. Then, the excess solvent was evaporated in a rotary evaporator at 55 °C under vacuum. As such, a white shiny crystalline MACl powder was obtained. The obtained powder was dissolved in ethanol (200 mL) by constant stirring at 40 °C for 2 h to purify the obtained powder. After complete dissolution, diethyl ether (200 mL) was added for precipitation. The precipitated powder was filtered out from the solution. This purification step was repeated twice. The obtained MACl powder was dried overnight in a vacuum oven at 60 °C.

In the second step, equimolar solutions of the obtained white MACl powder (1 M, 2.78 g) and PbCl_2_ (1 M, 0.58 g) were dissolved in DMSO (10 mL) by stirring at 60 °C. After complete dissolution, the solution was filtered through a 0.22 μm syringe filter into a crystallization dish. The dish was covered with aluminum foil, and a few holes were punched in the foil for slow evaporation resulting in better crystallization. The dish was then kept undisturbed at a constant temperature of 100 °C for 1–2 days. After 1–2 days, transparent MAPbCl_3_ crystals were obtained with approximate dimensions of 5 × 4 × 2 mm^3^. The crystals were then cleaned with acetone and dried overnight in a vacuum oven at 60 °C.

### 2.3. Characterization Techniques

A standard Raman spectrometer (LabRam HR800, Horiba Co., Longjumeau, France) was used to perform the Raman measurements. The single crystal was excited by using a diode-pumped solid-state laser with a wavelength of 532 nm. The probed frequency range was from 10 to 3500 cm^−1^. The Raman spectrometer was equipped with a low-frequency notch filter, by which the lowest frequency limit could be 10 cm^−1^. All measurements were taken by using an optical microscope (BX41, Olympus Co., Tokyo, Japan) with a 50-magnification objective lens at a backscattering geometry c(a, a+b)$\bar{c}$, where the *a*, *b*, and *c* denote the cubic axes. The scattered light is collected concurrently along the same path as that of the incident light in this geometry. A silicon standard sample with a single peak at 520 cm^−1^ was used to calibrate the Raman spectrometer before recording measurements. The measurements were performed at a temperature ranging from −196 °C to room temperature (RT) by using a cryostat (Linkam THMS600, Linkam Scientific Instruments Ltd., Surrey, UK) with a temperature stability of 0.1 °C. One and a half minutes of waiting time were given to reach the thermal equilibrium after the target temperature was achieved for every measurement. The intensity of all the measured Raman spectra was corrected by considering the Bose–Einstein thermal factor.

The powder XRD pattern was obtained in the 2θ angular range from 10 to 60° at RT by using a high-resolution XRD spectrometer (PANalytical; X’pert PRO MPD, Malvern, UK) at the Cu K-radiation (λ =1.5406 Å). For the measurements, single crystals were crushed into crystalline powders. PANalytical Software (X’pert highscore v1.1) was used to analyze the XRD patterns.

The PL spectrum was measured using a PL spectrometer (LabRam HR800, Horiba Co., Longjumeau, France) at RT. The slit width was 200 μm, and a diode laser at 375 nm was used as an excitation source. In addition, transmission measurements were taken using an optical absorption spectrometer (Duetta, Horiba Instruments, Kyoto, Japan) with a slit width of 100 μm where a tungsten–halogen lamp of 1 kW power was used as a light source.

A standard tandem multi-pass Fabry–Perot interferometer (TFP-2, JRS Co., Zürich, Switzerland) was used to record the Brillouin spectra with an excitation source of a 532 nm wavelength. Backscattering geometry was employed for the measurement by using a modified microscope (BH-2, Olympus, Tokyo, Japan). The same temperature stage as that used for the Raman experiment was used to control the temperature.

## 3. Results and Discussions

### 3.1. Structural Phase Transitions Probed by Raman Spectroscopy

Figure 2a shows the photo of the grown MAPbCl_3_ single crystal, and the unit cell of the cubic MAPbCl_3_ at RT is shown in Figure 2b, where the MA cation is located inside the octahedral spaces, i.e., between octahedral units, while Cl is present at the octahedral corner. Figure 2c shows the powder XRD pattern of the MAPbCl_3_ single crystal measured at RT. The diffraction peaks matched well with the indices of the cubic phase, and no extra peaks were observed, confirming the correctly synthesized composition of the single crystal. The sharp and distinct diffraction peaks proved the crystalline nature of the sample. The lattice constant obtained from the XRD pattern is 5.67 Å. The Goldschmidt tolerance factor of MAPbCl_3_ is close to unity (T ≈ 0.93) which justifies the cubic structure at RT.

Figure 3 shows the absorption and PL spectra of an MAPbCl_3_ single crystal measured at RT. The PL peak was observed at 406 nm, and the spectrum exhibits an asymmetrical line shape as previously reported [15]. Moreover, the peak position is blue shifted compared to first exciton peak which might be due to the existence of shallow-level traps between the band edges [28]. The absorption spectrum showed an absorption edge at 423 nm corresponding to an optical bandgap of 2.93 eV obtained using Tauc plot’s method, as shown in the inset of Figure 3. This is consistent with previous reports [19,29].

Temperature-dependent Raman spectroscopy was used to investigate the optical phonons modes’ behaviors of MAPbCl_3_ in order to characterize the structural phase transitions and to determine their exact temperature. Figure 4a–c show the temperature-dependent Raman spectra in the frequency range from 10 to 3500 cm^−1^ in a wide temperature range from −190 °C to RT. At RT, MAPbCl_3_ is cubic. It transforms into tetragonal and orthorhombic phases subsequently as the temperature is decreased. In the RT centrosymmetric cubic phase, the Raman modes are inactive in principle. However, broad Raman modes were observed due to a random intrinsic disorder induced by the freely rotating MA units and their displacements. On the contrary, for the low-temperature orthorhombic phase, MAPbCl_3_ undergoes global symmetry breaking caused by the tilting of the PbCl_6_ octahedra resulting in many distinct Raman modes.

The crystal structure of MAPbCl_3_ consists of PbCl_6_ octahedra and MA cations located in the octahedral spaces. Hence, vibrational modes related to PbCl_6_ octahedra, MA cation motions, and internal modes of the MA cation can be observed in different frequency ranges in the Raman spectra. We classified the Raman spectra and the relevant modes into three regions as shown in Figure 4. The first region below 400 cm^−1^ contains Pb–Cl octahedral vibrational modes and other modes related to the translational motions of the crystal lattice. The second region (400~1600 cm^−1^) includes the restricted torsional mode and rocking modes of the MA cation along with the bending modes of CH_3_ and NH_3_. The third region above 2700 cm^−1^ consisted of internal vibrational modes of the MA cation, such as stretching vibrational modes of CH_3_ and NH_3_. The Raman spectra for the three regions displayed in Figure 4a–c show clear changes upon temperature variation. The low-temperature spectrum in the orthorhombic phase (Figure 4a) exhibits several distinct and well-resolved peaks that broaden and coalesce as temperature increases. The Raman spectra in the mid-frequency region and the high-frequency region shown in Figure 4b,c display several changes in the peak positions and widths of the Raman modes upon temperature change. Individual mode behaviors and transition temperatures will be revealed through the curve-fitting analysis of these modes.

Anomalous changes in Raman shifts and full width at half maximum (FWHM) reveal important information related to the phase transition. Fitting analyses were performed to obtain the exact Raman shifts and FWHM of each Raman mode in the orthorhombic phase using the Lorentzian function. Due to uniformly dispersed dynamic disorder in the lattice caused by the onset of restricted motions of MA cations, the Raman peaks are expected to widen as the temperature rises [23]. The Lorentzian line shape was used to fit all the Raman spectra after the Bose–Einstein correction, as represented by the following relation:$$I_{R}\left(\nu \right)=I\left(\nu \right)∕\left[n\left(v\right)+1\right]$$
where the corrected Raman intensity $I_{R}\left(\nu \right)$ was obtained from the measured Raman intensity $I\left(\nu \right)$ using the Bose–Einstein thermal factor $n\left(v\right)={\left[\exp\left(\frac{h\nu }{kT}\right)-1\right]}^{-1}$. In the final expression, *h* and *k* are the Planck’s constant and Boltzmann constant, respectively, while *ν* and *T* are the frequency and the absolute temperature, respectively. The mode assignment was performed for the Raman spectrum observed at −196 °C because the Raman modes are well resolved at low temperatures due to smaller damping factors.

Table 2 shows the mode assignment for all the Raman modes of MAPbCl_3_. A total of 36 modes were observed in a wide frequency range from 10 to 3500 cm^−1^ in the orthorhombic phase. The lattice modes were observed from 10 to 193 cm^−1,^ whereas individual MA cation modes were present at higher frequencies. In the low-frequency range, modes were observed at 26, 42, 55, 60, 68, and 77 cm^−1^. The lowest frequency mode in MAPbCl_3_ was observed at 26 cm^−1^, while in FAPbCl_3_, the lowest frequency mode was observed at 37 cm^−1^ [30]. This shift in mode frequency might be due to the increased cationic radius of FA^+^ and a comparatively softer lattice of MAPbCl_3_ [17,30]. Leguy et al. observed low-frequency modes at 42, 54, 61, and 75 cm^−1^ associated with octahedral twisting and distortion [23]. In our results, the modes at 26 and 68 cm^−1^ were newly observed compared to their study. On the other hand, the mode at 26 cm^−1^ was observed by another report where it was associated with the motions of an MA cation [20]. However, we assign this mode to the lattice librational mode based on its similarity (in terms of intensity and position) to a similar lattice mode in MHyPbCl_3_ (methylhydrazinium lead chloride) which shares the same PbCl_6_ octahedra as MAPbCl_3_ [17]. A mode at 68 cm^−1^ was newly observed in our Raman spectrum, which is absent in any former Raman spectroscopic studies. This mode is probably attributed to the octahedral distortion similar to the mode at 77 cm^−1^ with the degeneracy being lifted due to lower symmetry in the orthorhombic phase. The modes observed at 93–193 cm^−1^ were associated with Pb–Cl bending and stretching modes, as suggested in a previous report [25]. For the same frequency range, Leguy et al. indicated that these modes result from MA cation motions coupled to the Pb–Cl octahedral motions, as these modes would be absent if MA cations were solely present in a vacuum [23].

The torsional mode of the MA cation was present at 484 cm^−1^. This mode is very sensitive to the halide composition; for instance, it is found at 484 cm^−1^ in MAPbCl_3_, at 323 cm^−1^ in MAPbBr_3_, and at 249 cm^−1^ in MAPbI_3_ [25]. In addition, this mode is also affected by the change in A-site cations [31]. Similar behavior was observed for the torsional mode of the FA cation in FA-based halide perovskites [30]. This shifting trend suggests a strong interaction between the organic cation and the halide atoms. The MA rocking modes were present at 923 and 1265 cm^−1^. The C–N stretching mode was found at 976 cm^−1^ and the symmetric and asymmetric bending modes of CH_3_ and NH_3_ were located in a range from 1400 to 1600 cm^−1^. The Raman peaks observed at high frequencies from 3000 to 3200 cm^−1^ include those modes associated with the symmetric and asymmetric stretching of CH_3_ and NH_3_. The assignment of the observed modes is shown in Table 2 based on previous reports [17,20,23,25]. Three peaks located at 68, 2920, and 3201 cm^−1^ could not be found in any of previous reports, and hence were not assigned to any vibrational mode.

In MAPbCl_3_, all three phases have a different lattice size and symmetry, thus the Raman modes for each phase are expected to be distinct with a different number of allowed modes from each other. The space group of the cubic phase is $Pm\bar{3}m$. It is P4/mmm for the tetragonal phase. Regarding the orthorhombic phase, $P222_{1}$ was suggested earlier [24], however, more recently, Pnma was proposed [32]. Figure 5a–c represent the low-frequency Raman spectra observed below 400 cm^−1^ along with the Lorentzian fitting curves in the orthorhombic, tetragonal, and cubic phases of MAPbCl_3_, respectively. The change in the Raman spectra is drastic, confirming the existence of three different phases in MAPbCl_3_, as has also been confirmed in previous reports [20]. In the low-temperature orthorhombic phase (Figure 5a), several sharp, resolved peaks can be clearly identified. Nevertheless, the peaks combine to form a rather broad spectrum in the tetragonal phase (Figure 5b). In the cubic phase (Figure 5c), many peaks disappear, and a broad spectral feature remains.

Raman shifts and FWHM were plotted as a function of temperature to identify the phase transition temperatures, as shown in Figure 6, Figure 7 and Figure 8 and Appendix A. All Raman modes show clear anomalies near −114 °C and −110 °C which are related to the orthorhombic-to-tetragonal and tetragonal-to-cubic phase transitions, respectively. As it can be noted that the tetragonal phase is only stable for a narrow temperature range of ~4 °C, some slight discrepancies might occur in these phase transition temperatures depending upon the type of measurement technique. A very recent Raman spectroscopic study also reported the tetragonal to orthorhombic phase transition temperature to be at 160 K (−113 °C), which occurs at a minimal difference from our results [17]. Other reported values of transition temperatures using various techniques are summarized in Table 1. For convenience, the transition temperature from orthorhombic to tetragonal will be called *T*_1_ and that from tetragonal to cubic will be called *T*_2_.

First, we analyze all the lattice modes below 400 cm^−1^ that exhibit substantial changes depending on the crystal symmetry. The first and the lowest frequency mode was observed at 26 cm^−1^, which shows a step-like change at *T*_2_. Similar behavior is shown by the mode at 42 cm^−1^ (octahedra twisting), which first shifts to a higher wavenumber at *T*_2_ and then exhibits a red shift till RT. Some of the Raman modes, such as at 56 cm^−1^ (octahedral twisting mode) and 119 cm^−1^ (asymmetric bending mode of Cl–Pb–Cl), show a red shift upon heating and finally disappear at *T*_2_. Furthermore, as the temperature is increased, the modes at 77 cm^−1^ (octahedral distortion mode), 144 cm^−1^ (symmetric stretching mode of Cl–Pb–Cl), and 193 cm^−1^ (symmetric stretching mode of Pb–Cl) disappear at *T*_1_. The mode at 93 cm^−1^ exists as a shoulder peak of the mode at 98 cm^−1^ (symmetric bending mode of Cl–Pb–Cl) that shows a red shift and vanishes at *T*_1_. The mode at 98 cm^−1^ shows an anomaly near *T*_1_ and persists up to RT with a weak blue shift. The mode at 164 cm^−1^ (asymmetric stretching mode of Pb–Cl) shows a weak red shift until *T*_2_ where a slight anomaly occurs and, then, continues to show a blue shift until RT. In the tetragonal and cubic phases above *T*_1_, a new mode appears at 237 cm^−1^. The appearance of this mode in the tetragonal phase is consistent with an earlier report where they observed a similar mode at 235 cm^−1^ and associated it with the rotational motion of the MA cation [20]. Furthermore, the previous study revealed that this mode does not persist in the cubic phase up to 300 K. Similarly, we could see that this mode is seen up to −50 °C and then disappears. The origin of this mode might be due to the unlocking of the MA cation that allows many possible reorientations causing the appearance of new modes in the spectra with a broad spectral feature. Another very recent Raman study revealed that the low-frequency lattice modes of MAPbBr_3_ display very pronounced anomalies at transition temperatures [33].

The FWHMs of these modes, which are shown in Appendix A, also display clear anomalies at the two transition temperatures. The FWHMs are very low in the orthorhombic phase while they become much larger in the high-temperature cubic phase. This indicates that the ordered arrangement of MA cations in the orthorhombic phase is responsible for weak anharmonicity and the low damping of optic phonons. A recent Raman study revealed that the MA cations are ordered along a specific crystallographic direction in the orthorhombic phase [34]. On the other hand, the C–N axis of the MA cations is disordered along equivalent <110> directions in the cubic phase [21]. Free rotating dynamics of MA cations in the cubic phase induce strong anharmonic lattice interactions causing the phonon damping larger, as indeed seen in the FWHM data. Some modes show strong broadening upon heating until they disappear at a particular transition temperature. For instance, the mode at 77 cm^−1^ (octahedra distortion mode) first significantly broadens and then disappears at *T*_1_. A previous study revealed that the FWHM of a typical cage mode of MAPbCl_3_ can reach ~40 cm^−1^ [23], however, in our Raman spectra, an average width of ~20 cm^−1^ was observed, which is similar to the reported values in other halide perovskites such as MAPbBr_3_ (~15 cm^−1^) and MAPbI_3_ (~10 cm^−1^) [23]. All low-frequency lattice modes in the cubic phase display very broad spectral features which are attributed to the disorder effect and heterogeneous local environment caused by rotatable MA cations.

The mid-frequency region ranging from 400 to 1600 cm^−1^ mainly includes the internal modes related to MA cation, for instance, torsional, rocking, and bending vibrational modes. The torsional mode (τ) of the MA cation is located at 484 cm^−1^. Several other studies observed this mode at 483 [23], 484 [17], and 488 cm^−1^ [25]. Since the MA cation’s movement is related to the inorganic cage via NH–X hydrogen bonding, altering the halide atom has a significant impact on this mode, for instance, from 249 cm^−1^ (MAPbI_3_) to 488 cm^−1^ (MAPbCl_3_) [25]. This mode experiences a significant step-like hardening at *T*_1_ and a minor shift at *T*_2_ upon increasing temperature and then persists up to RT. The mode broadening (see Appendix A where the FWHMs of all modes in the intermediate frequency range are shown) upon heating may be ascribed to the weakening of hydrogen bonding between the MA^+^ cation and the halogen Cl^-^ in the high symmetry cubic phase consistent with previous studies [17,20]. The FWHM of the torsional mode is the highest among other high-intensity modes at low temperatures, indicating that this mode is very anharmonic and sensitive to the local environment. Similar behavior was observed for other bromine and iodine-based MA halide perovskites [23].

The mode at 923 cm^−1^ is related to the first rocking mode (ρ) of the MA cation. This mode shows splitting in the low-temperature range from −196 °C to −160 °C and then merges into one mode. This mode shows a step-like anomaly near *T*_1_ where the Raman shift suddenly increases. A similar anomalous rise in the Raman shift is shown by the C–N stretching mode (*ν*) present at 976 cm^−1^. An increase in the mode frequency at *T*_1_ reflects the change in the C–N bond strength. This result is consistent with the changes observed from MAPbBr_3_ [35], where the Raman shifts of the rocking mode and the C–N stretching mode suddenly increased when it passed through *T*_1_. This indicates the weakening and lengthening of the C–N bond in the orthorhombic phase due to the new hydrogen-bond configurations in the orthorhombic phase [36]. However, the second rocking mode located at 1265 cm^−1^ shows an opposite temperature dependence where the frequency decreases as the temperature is increased with a small anomaly at *T*_1_. This might indicate that the second rocking mode does not involve any stretching of the C–N bond; it is rather related to the rocking of the C–N bond in a rigid way [36]. The same temperature dependences of the rocking and C–N stretching modes of MAPbBr_3_ and MAPbCl_3_ suggest that the temperature evolution of the H–X (X = Cl or Br) bonds is nearly the same in these two systems.

It is interesting to note that the torsional and the first rocking mode of the MA cation significantly broadens across the phase transitions. This can be attributed to the fact that the orientational degrees of freedom of the MA cation increase as the crystal moves from the low-symmetry orthorhombic phase to the high-symmetry cubic phase, resulting in different surrounding environments for the MA cation inside the lattice. At high temperatures, the dynamic disorder of the MA cations allows heterogeneous environments letting different mode frequencies appear together, which forms a broad peak instead of multiple peaks due to their superposition, high damping, and weak intensity. Another possibility may be that the size of the octahedral space is reduced in the cubic phase, which increases the steric hindrance effect and, hence, increases dynamic coupling causing the width to increase [23].

The symmetric and asymmetric bending modes (δ_s/as_) of CH_3_ are located at 1421 and 1457 cm^−1^, respectively. Both modes show hardening, i.e., a blue shift near *T*_1_ upon heating. The symmetric and asymmetric bending modes (δ_s/as_) of NH_3_ are located at 1473 and 1596 cm^−1^, respectively. The temperature dependences of the Raman shift of all these modes are consistent with a previous report [17]. The mode frequency at 1473 cm^−1^ sharply increases at *T*_1_ while the mode at 1596 cm^−1^ shows a red-shift behavior. However, in our study, both main modes have new shoulder peaks at 1467 and 1602 cm^−1^, which disappear at *T*_1_ and *T*_2_, respectively, as the temperature is increased. This seems to reflect the low-symmetry environment of the orthorhombic phase resulting in the lift of degeneracy. Interestingly, such mode splitting is more common in Cl-based halide perovskites according to theoretical analyses [20,37]. The intensities of several mode change across transition temperatures. For instance, the intensities of the *ν* (C−N) and δ_s_ (NH_3_) mode significantly decrease as the crystal moves from the orthorhombic to the cubic phase. This implies that the orthorhombic phase is more ordered than the cubic phase from the viewpoint of cation dynamics and, thus, well-defined strong Raman modes are observed.

Figure 8a–c show the temperature dependence of the Raman shifts of the high-frequency modes above 2700 cm^−1^. The modes in this frequency range are associated with the CH_3_ and NH_3_ symmetric/asymmetric stretching (*ν*_s/as_) vibrations. The mode at 2822 cm^−1^ with a shoulder peak at 2819 cm^−1^ is associated with the asymmetric vibration of the C–H bond. The shoulder peak disappears at *T*_1_ while the 2822 cm^−1^ mode shows a sudden increase in the Raman shift. The mode at 2900 cm^−1^ does not show a noticeable anomaly except for slight softening near *T*_1_ upon heating. The mode at 2920 cm^−1^ continues to increase until *T*_1_ and disappears. For some modes in this range, discrepancies exist for the mode assignments; for instance, Nguyen et al. mentioned that modes in the 2800–2960 cm^−1^ range are originated from MA vibrations in phase-separated MACl present at the surface of the crystal in a nanoscopic scale [17]. The overall trend of temperature-dependent Raman shifts in our results is nearly the same as that reported by Nguyen et al. However, in this work, we adopt the mode assignment proposed by Leguy et al. who associated these modes with the C–H and N–H symmetric/asymmetric stretching modes [23]. The FWHMs of these modes shown in Appendix A also exhibit clear anomalies at the transition temperatures. The clear and noticeable anomalies shown by these modes near the phase transition temperatures demonstrate that these modes must be related to the bulk properties, i.e., the internal organic–inorganic structure and thus the structural phase transitions.

The CH_3_ asymmetric vibrations are present at 2967/2971 and 3032 cm^−1^ in our results. The modes at 2967 cm^−1^ and 2971 cm^−1^ are split and merged at *T*_1_ with an increasing temperature, and, persist up to RT. A similar mode was observed by Nguyen et al. (at 2957 cm^−1^ in the experiment and 2974 cm^−1^ via theoretical calculations) which they associated with the MA vibration in MACl [17]. However, our results revealed that they are directly correlated with the structural phase transition and, thus, can be assigned to the CH_3_ asymmetric vibrations in the bulk MAPbCl_3_ single crystal consistent with the mode assignment by Niemann et al. [25]. Furthermore, the CH_3_ symmetric vibration was found at 3032 cm^−1^ with a shoulder peak at 3040 cm^−1^ which disappears at *T*_1_ while the principal peak shows a strong increase in Raman shift at *T*_1_ and blue shifts until RT. This is evidence that these high-frequency modes are associated with the bulk vibrational properties rather than with surfaces having a composition different from that of the bulk.

The symmetric/asymmetric stretching modes of NH_3_ are located at 3109, 3140, 3156, and 3180 cm^−1^ As shown in Figure 8c, the symmetric stretching mode of NH_3_ at 3109 cm^−1^ exhibits a blue shift as the temperature increases. The second symmetric stretching mode of NH_3_ at 3180 cm^−1^ shows a drastic increase in Raman shift at *T_1_* and then persists to RT. The asymmetric stretching mode of NH_3_ at 3140 cm^−1^ disappears at approximately −170 °C and the 3156 cm^−1^ mode disappears near *T*_1_. The significant frequency shifts for all CH_3_ and NH_3_ modes imply that the hydrogen bond strength changes with the temperature change, especially at temperatures near the transition temperatures.

Orthorhombic-to-tetragonal phase transition was found at −114 °C while tetragonal to cubic phase transition was found at −110 °C, both of which were of a first-order phase transition according to a previous report [17]. The Raman shifts and FWHM of nearly all modes are significantly affected by the phase transitions. The phase transitions of MAPbCl_3_ were characterized by three main anomalies. Firstly, the temperature dependences of low-frequency lattice modes were revealed for the first time. No soft mode was observed, suggesting that the present system is not a displacive type or that the soft mode is located at wavevectors other than the zone center in the first Brillouin zone. In addition, the low-frequency central mode, which appeared in MAPbBr_3_ [36], was not observed, indicating that the effect of MA off-centering, which was supposed to be responsible for the central mode, plays a different role in the light-scattering spectrum of MAPbCl_3_. Secondly, the hardening or softening of several modes, such as C–N stretching and MA rocking modes, of MAPbCl_3_ at the transition temperatures were nearly the same as those of MAPbBr_3_ [36]. In addition, anomalies of the low-frequency lattice modes were very similar to the case of MAPbBr_3_ [33]. This suggests that the H–halogen interactions, MA freezing/unlocking, and their changes across the transition point are very similar between MAPbCl_3_ and MAPbBr_3_. Finally, the half widths, which are inversely proportional to the phonon lifetimes, change drastically across the phase transition. In all modes, the half widths are very small in the orthorhombic phase and then discontinuously increase upon passing through the phase transition points. The alignment of MA cations along a specific crystallographic direction in the orthorhombic phase is transformed into a state of dynamic disorder that unlocks their motions in the cavities, resulting in high-phonon damping caused by high anharmonicity and a local heterogeneous environment. Nearly the same damping behaviors were recently reported for the MAPbBr_3_ system [33]. Finally, we need to mention that the phase transition temperatures and the temperature range of the tetragonal phase are significantly different depending on the research groups and experimental techniques. It would be interesting to investigate the effect of the impurity level on the stability region of the tetragonal phase in different types of samples, such as single crystals, thin films, etc.

### 3.2. The Degradation Process of MAPbCl_3_ Probed by Raman and Brillouin Scattering

Exposure of MAPbCl_3_ to high temperatures induces structural evolution and thermal decomposition. Tracking this process is important for practical device applications. Raman spectra were measured at high temperatures from RT to 200 °C (See Appendix A). The Raman spectra were nearly the same from RT up to 180 °C. However, at 200 °C, the spectra changed drastically, indicating the thermal decomposition of the material. This temperature is consistent with the thermogravimetric analysis which is shown in Appendix A. Another previous study also reported 200 °C as the onset of decomposition [38]. In the Raman spectra at 200 °C, the high-frequency modes disappeared, and the low-frequency modes drastically changed. Moreover, the appearance of the crystal significantly changed from transparent to a white shade at 200 °C (Appendix A). The color remained white upon cooling down the crystal to RT, indicating that it is irreversibly changed and permanently decomposed. The Raman spectrum of the degraded sample measured at RT, which is described by the black, solid line in Figure 9, is substantially different from that of the fresh sample.

To check the effect of time evolution on the structural change of MAPbCl_3_ and compare it with the temperature evolution described above, we kept the sample in the ambient condition for one month and then measured its Raman spectrum. The Raman spectrum, which is shown as a green, solid line in Figure 9, was different from the original spectrum of the pristine MAPbCl_3_ and was similar to that of the sample degraded by high temperatures. It would thus be interesting to compare the Raman spectra of possible byproducts and the degraded/decomposed MAPbCl_3_. Principally, MAPbCl_3_ should decompose into its two expected byproducts, namely PbCl_2_ and MACl. To check this, we compared the Raman spectra of these two expected byproducts with that of degraded MAPbCl_3_. Figure 9 shows a comparison of the Raman spectra of the byproducts and those of the two kinds of degraded MAPbCl_3_. Both spectra of two degraded samples showed that the low-frequency modes changed significantly and that the high-frequency modes vanished compared to that of the fresh MAPbCl_3_ sample. The Raman spectra of the two degraded MAPbCl_3_ samples are nearly the same as that of PbCl_2_ clearly indicating that part of the MAPbCl_3_ has been decomposed into PbCl_2_. The absence of similarity with the spectrum of MACl might be due to the weak MACl modes hidden under the strong Pb–Cl modes. Thus, we can conclude that both temperature-induced and prolonged time-induced degraded MAPbCl_3_ crystals are decomposed into PbCl_2_ with similar colors consistent with a previous report [39].

Brillouin spectroscopy can be used to probe low-frequency acoustic phonons which are sensitive to structural and chemical changes. A previous Brillouin scattering study revealed that acoustic phonon behaviors were directly associated with low-temperature structural phase transitions [18]. A high-temperature Brillouin scattering experiment was performed to compare the acoustic behaviors of pristine and degraded MAPbCl_3_. Figure 10a shows the temperature dependence of the Brillouin spectrum of pristine MAPbCl_3_ from RT to 200 °C. The Brillouin spectrum shows a significant change when the sample passes through a specific temperature of approximately 200 °C, which is similar to the result of the Raman experiment where the Raman spectrum exhibits substantial changes at this temperature. Figure 10b shows the Brillouin spectrum of degraded MAPbCl_3_ that was cooled to RT from 200 °C. Pristine MAPbCl_3_ shows two Brillouin doublets corresponding to the longitudinal acoustic (LA) and the transverse acoustic (TA) modes appearing at ~25 and ~8 GHz, respectively. This is a typical Brillouin spectrum of MAPbCl_3_ at RT [18]. However, the degraded MAPbCl_3_ shown in Figure 10b is significantly different from the typical spectrum of the fresh sample and exhibits very broad spectral features. This broadened spectrum is typically observed from ceramics or powders in which the excitation light can undergo multiple reflections and refractions, resulting in a wide range of allowed momentum transfer. Figure 11a,b display the temperature dependence of the frequency and the FWHM of the LA and the TA mode, respectively. Both modes show abrupt changes at ~200 °C, which indicates that thermal decomposition begins at this temperature. The present result demonstrates that Brillouin scattering is a useful tool in monitoring the degradation process of halide perovskite materials. Appendix A shows the comparison of the Brillouin spectra of the fresh MAPbCl_3_ single crystal that kept for one month under ambient conditions. In contrast to that of the fresh sample, the resonance peaks became asymmetric and widened in the degraded one. In addition, two TA modes appeared at frequencies other than that of the original crystal. This suggests that prolonged exposure to ambient conditions and the resulting degradation process can also be investigated by Brillouin scattering.

The exact monitoring of the structural evolution including the degradation process is a key parameter to be considered in practical device applications. This study consisting of both Raman and Brillouin scattering will be helpful to develop a deeper understanding of the structural evolution of MAPbCl_3_ and will also provide valuable data for optimizing the device performance by choosing the correct operating temperatures.

## 4. Conclusions

The structural evolution of a halide perovskite MAPbCl_3_ single crystal during phase transitions and temporal- or temperature-induced degradation processes was studied by combined Brillouin and Raman scattering techniques. Raman scattering highlighted new low-frequency lattice modes whose temperature dependences displayed clear and discontinuous anomalies at transition temperatures, similar to the case of MAPbBr_3_. This was consistent with the similarity of C–N stretching and rocking modes between the two systems suggesting that the temperature evolution of the H–halogen interactions and ordering/disordering of MA cations across the transition point was very similar. The broadening of the line widths, which are related to the phonon lifetimes, were observed at the transition point upon heating for nearly all Raman modes, indicating that the unlocking of the aligned MA cations and the resulting dynamic disorder caused substantial anharmonicity in the lattice. High-temperature Raman and Brillouin scattering results showed significant changes at 200 °C. A comparison of the high-temperature Raman spectrum of MAPbCl_3_ with that of PbCl_2_ suggests that MAPbCl_3_ decomposes into PbCl_2_ at high temperatures above 200 °C. Prolonged exposure of the fresh sample to ambient conditions for one month induced similar spectral changes in the Raman and Brillouin spectra as those of the high-temperature results. These results show that the combination of Raman and Brillouin scattering techniques can be a useful tool in monitoring the degradation process of lead-based halide perovskites.

## Figures and Tables

**Figure 1 materials-15-08151-f001:**
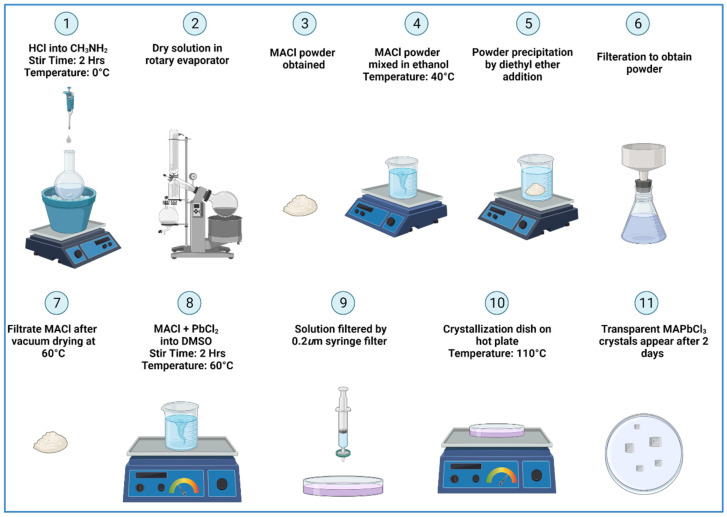
(Color online) A schematic representation of the MAPbCl_3_ single crystal growth process based on a solvent evaporation method (created with Biorender.com).

**Figure 2 materials-15-08151-f002:**
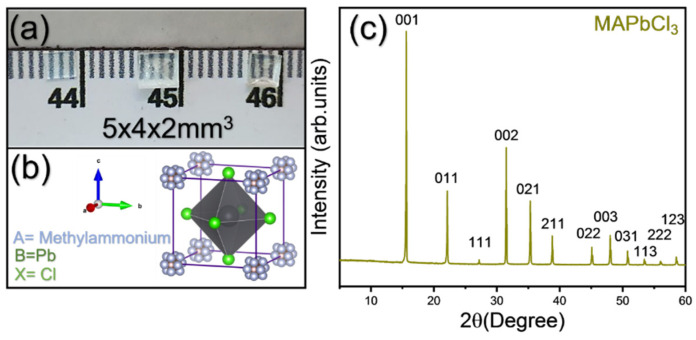
(Color online) A photograph of synthesized single crystals (**a**), a basic unit cell structure (**b**), and a powder X-ray diffraction pattern of MAPbCl_3_ in the cubic phase $Pm\bar{3}m$ (**c**).

**Figure 3 materials-15-08151-f003:**
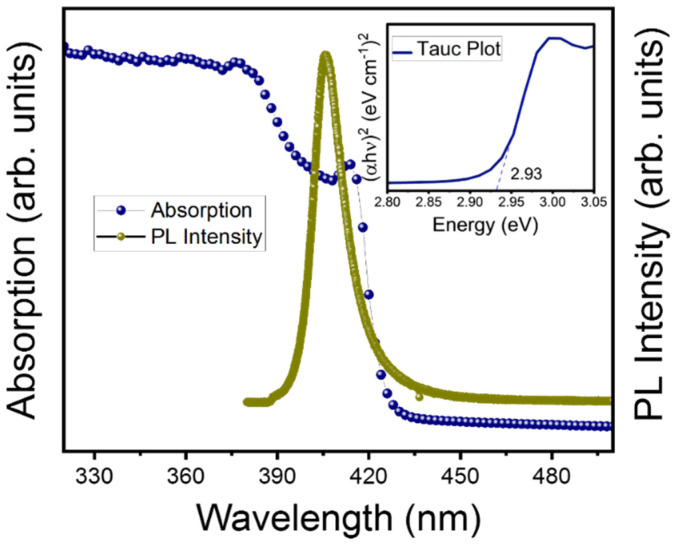
(Color online) Absorption and photoluminescence spectra of the MAPbCl_3_ single crystal in the cubic phase $Pm\bar{3}m$. Inset figure shows the calculated bandgap via the Tauc plot method.

**Figure 4 materials-15-08151-f004:**
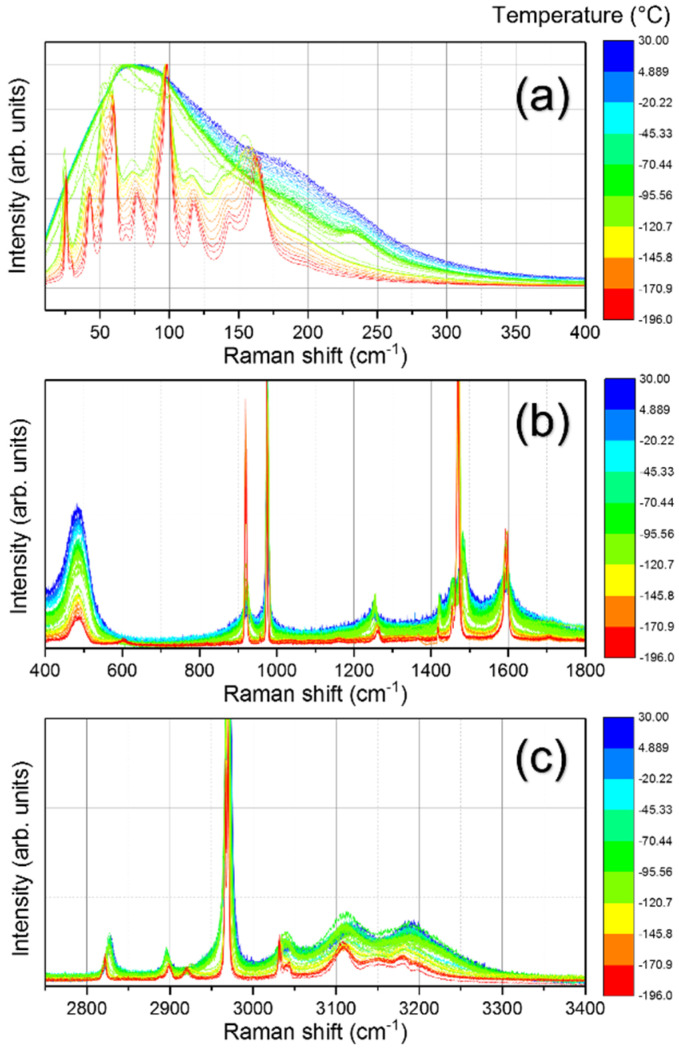
(Color online) Temperature-dependent Raman spectra of the MAPbCl_3_ single crystal in 10–400 cm^−1^ (**a**), 400–1800 cm^−1^ (**b**), and 2700–3400 cm^−1^ (**c**) frequency ranges from room temperature down to −196 °C.

**Figure 5 materials-15-08151-f005:**
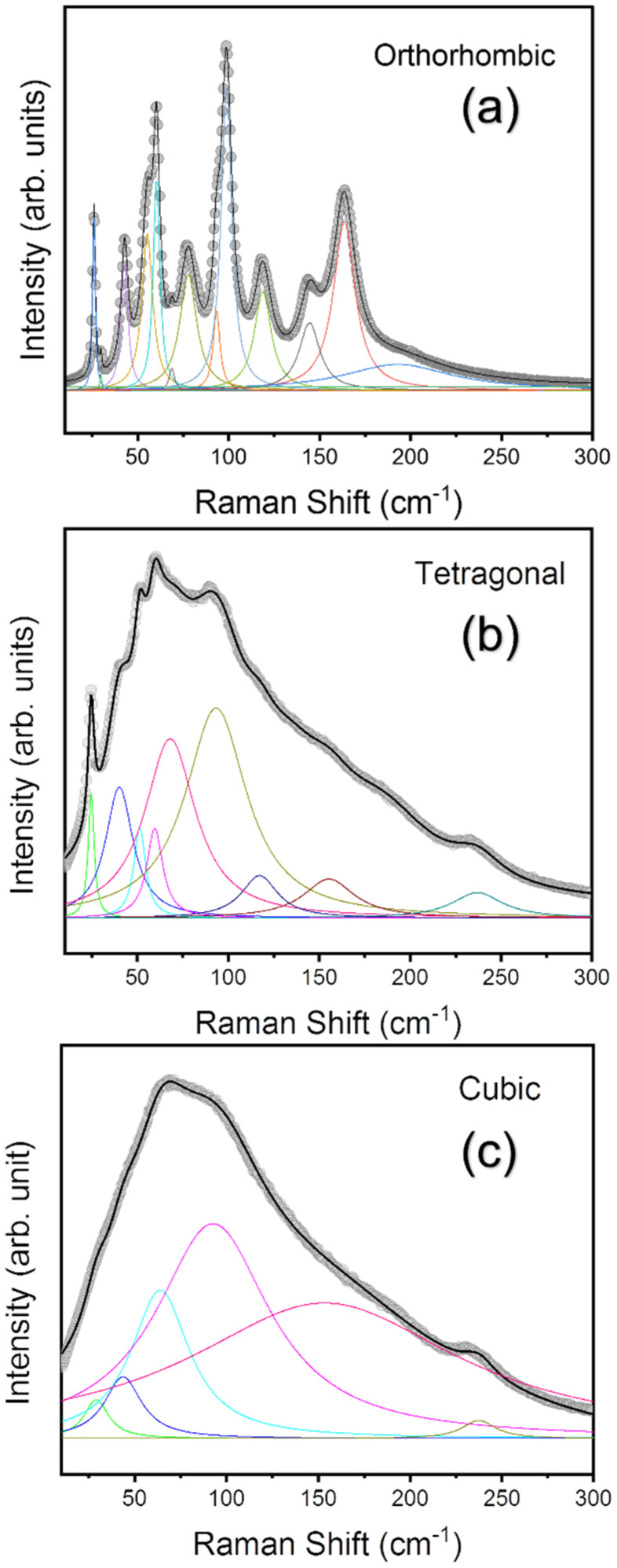
(Color online) Raman spectra and best-fitted curves of MAPbCl_3_ single crystal in the orthorhombic (**a**), tetragonal (**b**), and cubic phases (**c**). The grey circles and black solid line indicate raw and overall fitted spectrum, respectively. The colored solid lines represent fitting curves of individual Raman modes.

**Figure 6 materials-15-08151-f006:**
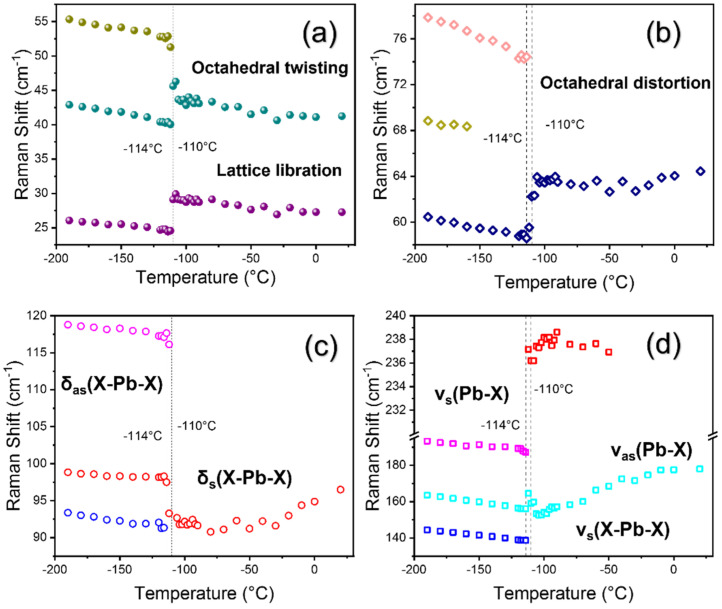
(Color online) Temperature dependences of the Raman shift in a 10–300 cm^−1^ frequency range (**a**–**d**). Phase boundaries are shown as vertical dotted lines along with the phase transition temperatures.

**Figure 7 materials-15-08151-f007:**
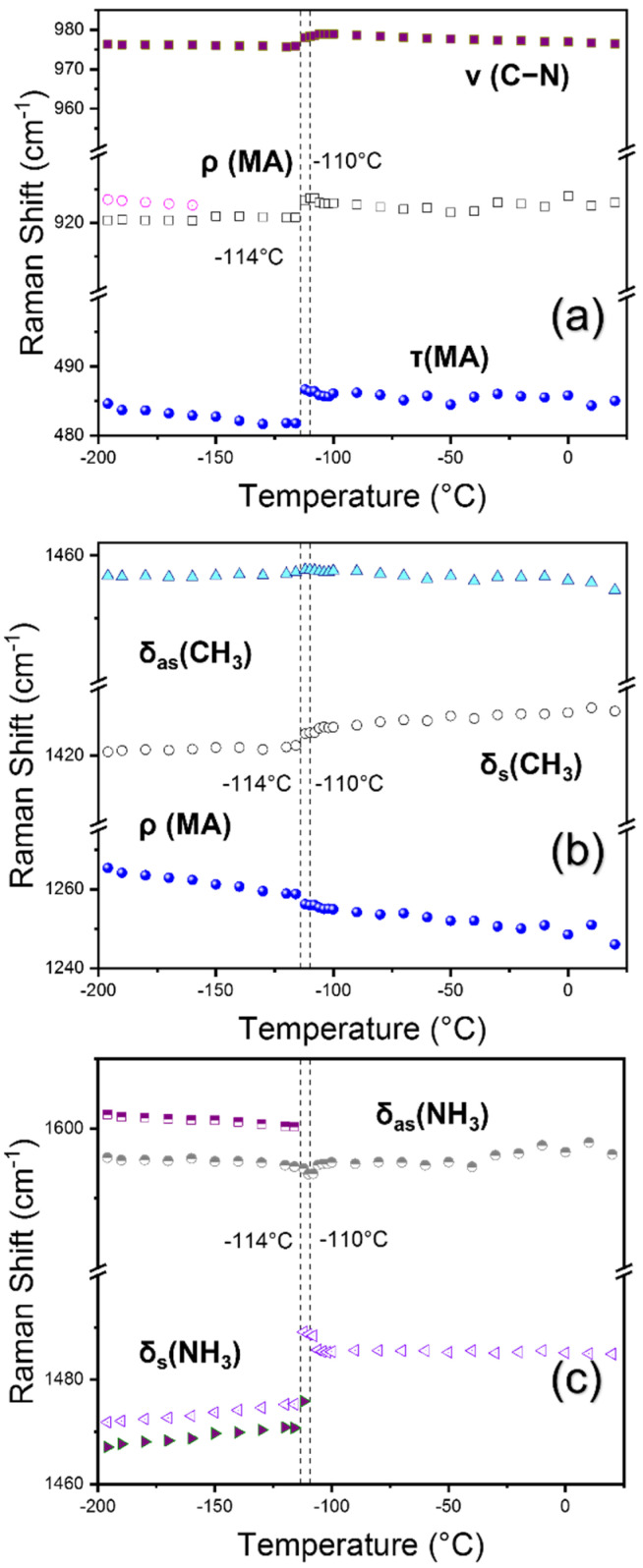
(Color online) Temperature dependences of the Raman shift in a 400–1700 cm^−1^ frequency range (**a**–**c**). Phase boundaries are shown as vertical dotted lines along with the phase transition temperatures.

**Figure 8 materials-15-08151-f008:**
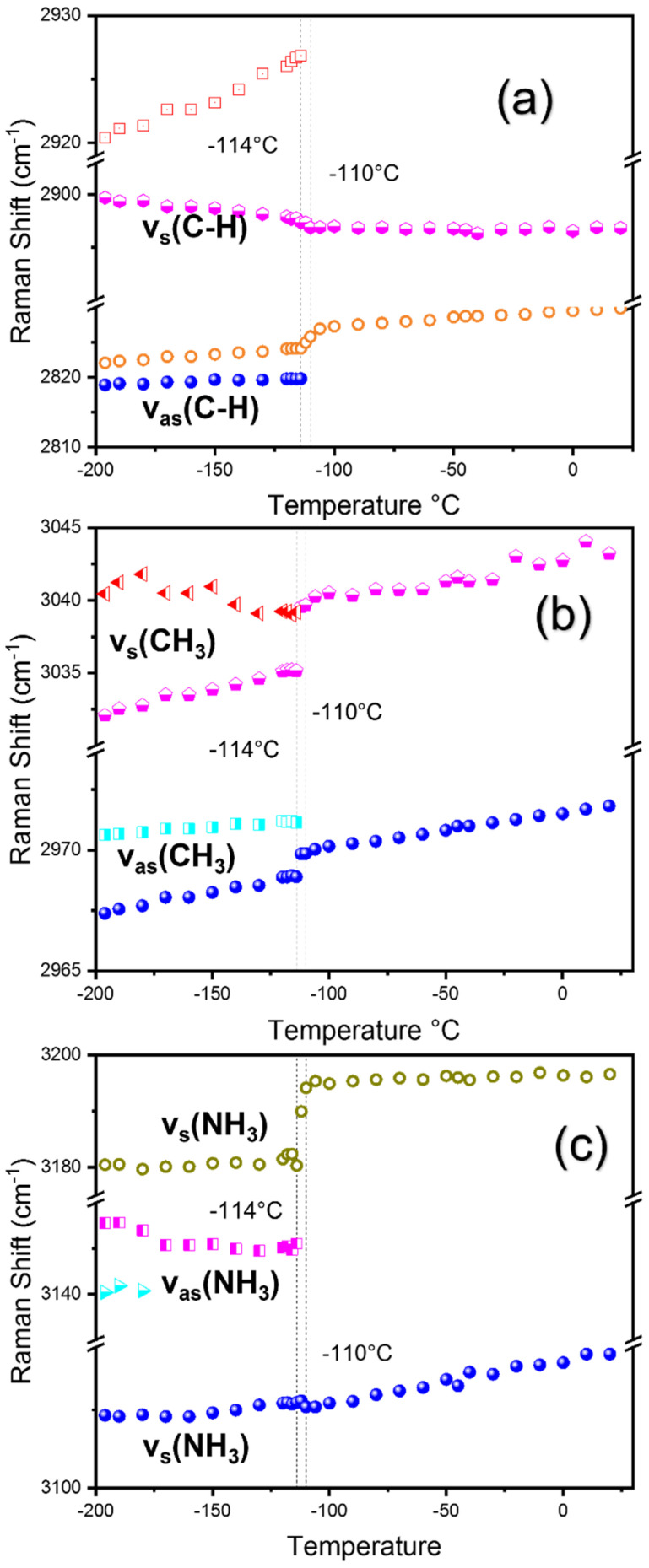
(Color online) Temperature dependences of the Raman shift in a 2700–3400 cm^−1^ frequency range (**a**–**c**). Phase boundaries are shown as vertical dotted lines along with the phase transition temperatures.

**Figure 9 materials-15-08151-f009:**
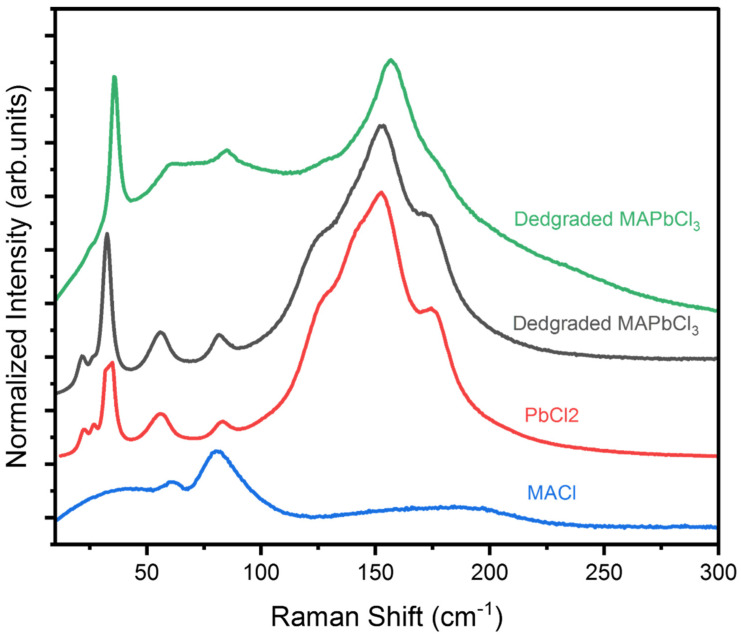
(Color online) Raman spectra of degraded MAPbCl_3_ compared with that of its byproducts (PbCl_2_ and MACl) at room temperature. The black, solid line indicates the crystal degraded due to exposure to high temperature and the green, solid line indicates the one exposed at ambient condition for a prolonged time. The Raman spectra of PbCl_2_ and MACl are also shown for comparison.

**Figure 10 materials-15-08151-f010:**
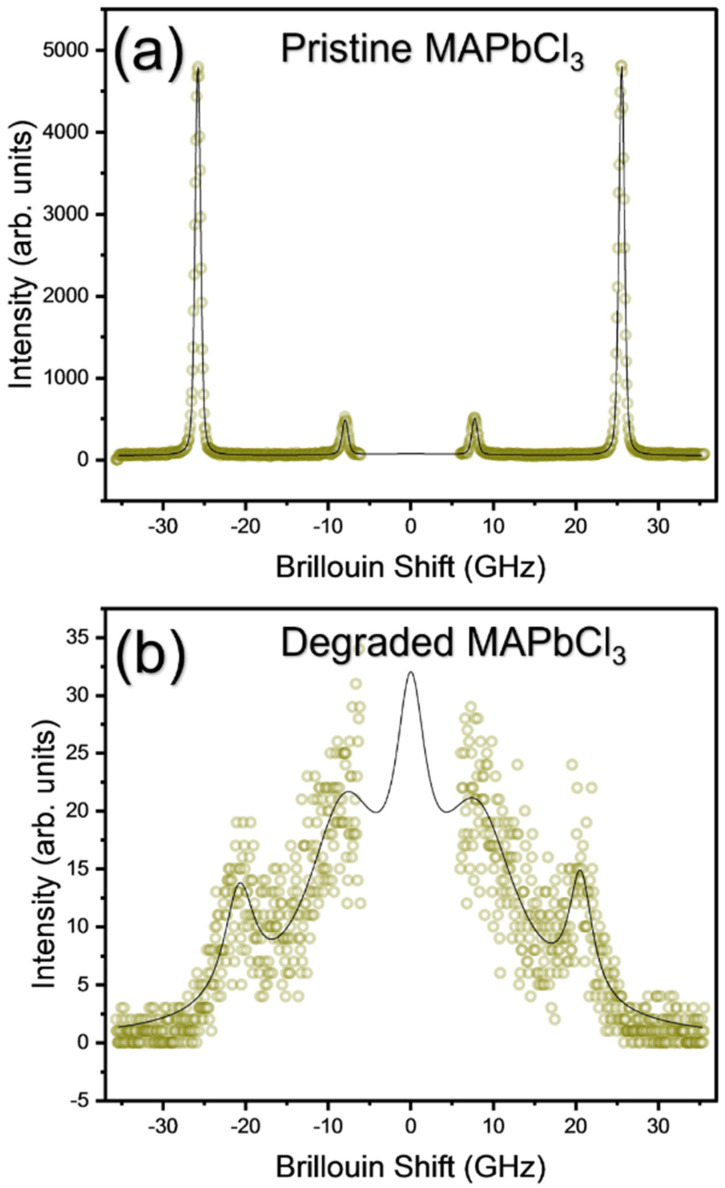
(Color online) Comparison of Brillouin spectra of pristine (**a**) and degraded MAPbCl_3_ at high temperatures and cooled to RT (**b**). The circles and the solid line indicate the raw data and best-fitted result, respectively.

**Figure 11 materials-15-08151-f011:**
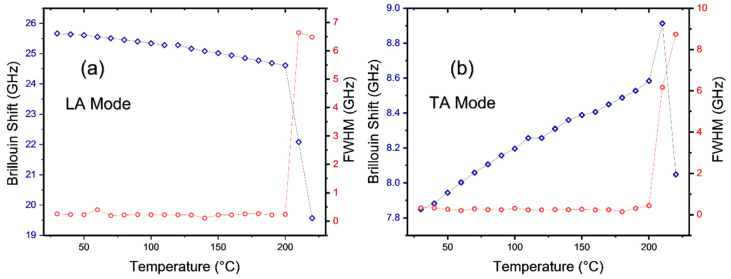
(Color online) Temperature dependence of the frequency (royal blue symbols) and the FWHM (velvet red symbols) of the LA (**a**) and the TA mode (**b**) measured by Brillouin scattering.

**Table 1 materials-15-08151-t001:** Comparison of phase transition temperatures of MAPbCl_3_ in previous studies.

Reports	Cubic to Tetra	Tetra to Ortho	Technique Used	Sample
Our work	167.2 K (−110 °C)	161 K (−114 °C)	Raman spectroscopy	Single crystal
Nguyen et al. [17]	Elusive	160 K (−113 °C)	Raman spectroscopy	Single crystal
Alvarez-Galván et al. [21]	167.5 K		X-ray diffraction	Crystalline powder
Onoda -Yamamur et al. [22]	177 K	171.5 K	Calorimetric and IR spectroscopy	Single crystal
Poglitsch et al. [24]	178.8 K	172.9 K	X-ray diffraction	Single crystal
Leguy et al. [23]	179 K	172 K	Raman and terahertz spectroscopy	Single crystal
Niemann et al. [25]	177.2 K	171.5 K	Raman and IR spectroscopy	Film
Hsu et al. [15]	185 K	175 K	X-ray diffraction	Single crystal

**Table 2 materials-15-08151-t002:** Raman mode assignments of vibrational modes in the 10–3500 cm^−1^ frequency range according to PbCl_6_ octahedra and MA cation molecular symmetry. All the vibrational modes correspond to the orthorhombic phase Pnma.

Mode	Assignment
26 s	Lattice libration [17]
29 vw	Octahedra twist/MA^+^ motion [20,23]
42 m	Octahedra twist/MA^+^ motion [20,23]
55 sh	Octahedra twist/PbCl_6_ motion [20,23]
60 vs	Octahedra twist/PbCl_6_ motion [20,23]
68 vw	
77 m	Octahedra distortion/PbCl_6_ motion [20,23]
93 sh	δ_s_ (Cl–Pb–Cl) [25]
98 vs	δ_s_ (Cl–Pb–Cl) [25]
119 m	δ_as_ (Cl–Pb–Cl) [5]
144 m	*ν*_s_ (Cl−Pb−Cl) [5]
164 s	*ν*_as_ (Pb–Cl) [25]
193 w	*ν*_s_ (Pb–Cl) [25]
237 w	R of MA+ cation [20]
484 m	τ (MA) [17]
920 s	ρ (MA) [17]
923 sh	ρ (MA)
976 vs	*ν* (C−N) [25]
1265 w	ρ (MA) [23,25]
1421 w	δ_s_ (CH_3_) [16,25]
1457 m	δ_as_ (CH_3_) [16,25]
1467 w	δ_s_ (NH_3_) [16]
1472 vs	δ_s_ (NH_3_) [25]
1596 m	δ_as_ (NH_3_) [23,25]
1602 m	δ_as_ (NH_3_)
2819 sh	*ν*_as_ (C–H) [23]
2822 w	*ν*_as_ (C–H) [23]
2900 w	*ν*_s_ (C–H) [23]
2920 w	
2967 sh	*ν*_as_ (CH_3_)
2971 vs	*ν*_as_ (CH_3_) [25]
3032 m	*ν*_s_ (CH_3_) [17]
3040 sh	*ν*_s_ (CH_3_)
3109 m	*ν*_s_ (NH_3_) [23]
3140 w	*ν*_as_ (NH_3_) [17]
3156 w	*ν*_as_ (NH_3_) [17]
3180 w	*ν*_s_ (NH_3_) [17]
3201 w	

δ: bending; ρ: rocking; *ν*: stretching; s/as: symmetric/asymmetric; τ: torsion; R: rotation; vs: very strong s: strong; m: medium; w: weak; vw: very weak; sh: shoulder.

## Data Availability

Data presented in this article is available on request from the corresponding author.

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
