# Peer review of "Structural Phase Transitions and Thermal Degradation Process of MAPbCl3 Single Crystals Studied by Raman and Brillouin Scattering"

_materials, 2022, doi:10.3390/ma15228151_

Round 1
Reviewer 1 Report
Jae-Hyeon Ko and coworkers reported an interesting work using Raman spectra to study the structural phase transitions and thermal degradation of MAPbCl3 perovskites. The work is well performed, and paper is well written. The novelty is high as well. However, there are still some problems in the experiments, references and introduction. Therefore, the following questions must be addressed.
1. The authors claimed that “MAPbCl3 shows a high bandgap energy of 2.88 eV and is, thus, transparent”. A reference that shows the MAPbCl3 bandgap of 2.88eV should be cited.
2. The introduction should be expanded and more discussion about the advantage/ applications of perovskites should be added. In this way, the significance of why study this material and the paper could be improved. For example, halide perovskites have been widely used in bioimaging (https://doi.org/10.1002/adfm.201604382), OLED and photocatalysis (https://doi.org/10.1021/acsmacrolett.0c00232), and the above related paper could be cited.
3. Figure 3, why is photoluminescence are blue shifted compared to the first exiticon peak? This is abnormal considering the stokes shift.
4. The experiments is figure 9 is well done. One more suggestion is the authors can do XRD for degraded MAPbCl3 to see if there is PbCl2’s peak. This can further prove the conclusion.
5. For experiment that “Prolonged exposure of the fresh sample to ambient condition”, one more factor need to consider is water. Small amount of water would usually cause phase change. This has been observed in many works but not in this work. Any reason?
Author Response
We appreciate the reviewers’ critical reading and valuable opinions on our manuscript. Comments raised by the two reviewers were considered and reflected in the revised manuscript as follows. We have written our response in blue color while the red color indicates the actual text changes done in the manuscript. The text already present in the manuscript is written in black color in our answers. In the original manuscript, changes were done using “Track changes: option in MS word. Moreover, in review section of MS word manuscript file, comments were added in black and blue color for reviewer 1 and reviewer 2 respectively.
<Response to Reviewer 1>
Jae-Hyeon Ko and coworkers reported an interesting work using Raman spectra to study the structural phase transitions and thermal degradation of MAPbCl3 perovskites. The work is well performed, and paper is well written. The novelty is high as well. However, there are still some problems in the experiments, references and introduction. Therefore, the following questions must be addressed.
- The authors claimed that “MAPbCl3 shows a high bandgap energy of 2.88 eV and is, thus, transparent”. A reference that shows the MAPbCl3 bandgap of 2.88eV should be cited.
- Thank you very much for the reviewer’s favorable opinion on our manuscript. We have inserted the reference for bandgap as suggested on page 2 (second paragraph, second line): “MAPbCl3 shows a high bandgap energy of 2.88 eV and is thus transparent, which is ideal for visible light spectroscopy [19].”
- The introduction should be expanded and more discussion about the advantage/ applications of perovskites should be added. In this way, the significance of why study this material and the paper could be improved. For example, halide perovskites have been widely used in bioimaging (https://doi.org/10.1002/adfm.201604382), OLED and photocatalysis (https://doi.org/10.1021/acsmacrolett.0c00232), and the above related paper could be cited.
- We expanded the introduction section by adding three more lines at the end of paragraph 2 of Introduction section indicated in red color. We also cited the very important requested articles. The changes in the manuscript were made as:
“Furthermore, LHPs are tempting technological materials to be used as photocatalysts [9] and as a probe in bioimaging field [10]. Particularly, methylammonium lead chloride (MAPbCl3) due to its wide band gap is famous for ultraviolet photodetection [11].”
The numbers for all other references were updated accordingly in the manuscript.
- Figure 3, why is photoluminescence are blue shifted compared to the first exiticon peak? This is abnormal considering the stokes shift.
- We inserted some explanation at page 5 (second paragraph, third line) as:
“Moreover, the peak position is blue shifted compared to first exciton peak which might be due to existence of shallow level traps between the band edges [28].”
- The experiments is figure 9 is well done. One more suggestion is the authors can do XRD for degraded MAPbCl3 to see if there is PbCl2’s peak. This can further prove the conclusion.
- We could not perform the XRD measurements due the limitation of time. However, we will try to use XRD in our future works as a secondary confirmative technique. Thank you for the nice suggestion.
- For experiment that “Prolonged exposure of the fresh sample to ambient condition”, one more factor need to consider is water. Small amount of water would usually cause phase change. This has been observed in many works but not in this work. Any reason?
- Thank you for your comment. We only considered degradation based on only two parameters (temperature and time). We did not particularly study the effect of humidity on the sample. This is an interesting topic and will be one of our future works.

Reviewer 2 Report
The authors combined Raman and Brillouin Scattering techniques to investigate vibrational properties of Organic-Inorganic Halide Perovskite MAPbCl3 in very wide range of frequencies as well as temperature. Therefore, the phase transitions were well identified. Most of results in this work are consistent with that found in previous works, which shows the reliability of the synthesis and characterization. However, there are few points should be improved, especially in presentation, as follows
1. More active tense should be used in the introduction, which shows authors proactively doing the research.
2. Caption of Figure 2 should mention that the information shown for cubic phase
3. The same for caption of Figure 3
4. The y-axis of Figure 4 (b) and (c) should be in the shorter range of intensity in order to help readers to see how temperature derives the signal. The Temperature scale and unit should be marked or mentioned in the Figure caption.
5. The caption of Table 2 should be more organized and more specific. The phases in which each peak appears should be shown . There is a typo for mode name, \rho was not explained white \tau was not Assignment row. Same things are in Figure 7 and 8.
6. There are three peaks 68, 2920 and 3201 cm-1 were not assigned or found in the previous works.
7. Paragraph 6, section 3.1 mentioned two different Orthorhombic phases were identified in the previous works. What's your Orthorhombic phase?
8. The region around tetragonal phase in Figure 6, 7, &8 should be zoomed in for clearer view. The same things for Figure S 1, 2, &3.
9. Figure 6 does not show mode symbols.
10. Caption of Figure 9 does not mention the temperature to measure the Raman spectral for PbCl2 and MACl.
11. The discussion and comparison with theoretical works (if available) might be helpful to determine the phonon modes. For instance, there have been two detailed calculations for MAPbI3 [PHYSICAL REVIEW B 92, 144308 (2015)], [J. Phys. Chem. C 122, 21703 (2018)]
Author Response
Response Letter (materials-1981969)
We appreciate the reviewers’ critical reading and valuable opinions on our manuscript. Comments raised by the two reviewers were considered and reflected in the revised manuscript as follows. We have written our response in blue color while the red color indicates the actual text changes done in the manuscript. The text already present in the manuscript is written in black color in our answers. In the original manuscript, changes were done using “Track changes: option in MS word. Moreover, in review section of MS word manuscript file, comments were added in black and blue color for reviewer 1 and reviewer 2 respectively.
<Response to Reviewer 2>
The authors combined Raman and Brillouin Scattering techniques to investigate vibrational properties of Organic-Inorganic Halide Perovskite MAPbCl3 in very wide range of frequencies as well as temperature. Therefore, the phase transitions were well identified. Most of results in this work are consistent with that found in previous works, which shows the reliability of the synthesis and characterization. However, there are few points should be improved, especially in presentation, as follows
- More active tense should be used in the introduction, which shows authors proactively doing the research.
- Thank you very much for the reviewer’s favorable opinion on our manuscript. We read the introduction section; we slightly modified the sentences in the introduction according to this comment.
- Caption of Figure 2 should mention that the information shown for cubic phase
- The caption of figure 2 was updated as:
“A photograph of synthesized single crystals (a), a basic unit cell structure (b), and a powder X-ray diffraction pattern of MAPbCl3 in the cubic phase .”
- The same for caption of Figure 3
- The caption of figure 3 was updated as:
“Absorption and photoluminescence spectra of the MAPbCl3 single crystal in the cubic phase . Inset figure shows the calculated bandgap via the Tauc plot method.”
- The y-axis of Figure 4 (b) and (c) should be in the shorter range of intensity in order to help readers to see how temperature derives the signal. The Temperature scale and unit should be marked or mentioned in the Figure caption.
- The y-axis of figure 4 (b) and (c) were shortened in intensity range. Moreover, the temperature scale title and unit were added in the figure at top right corner and temperature scale was updated in the figure caption as:
“Temperature-dependent Raman spectra of the MAPbCl3 single crystal in 10-400 cm-1 (a), 400-1800 cm-1 (b), and 2700-3400 cm-1 (c) frequency ranges from room temperature down to -196°C.”
- The caption of Table 2 should be more organized and more specific. The phases in which each peak appears should be shown . There is a typo for mode name, \rho was not explained white \tau was not Assignment row. Same things are in Figure 7 and 8.
- The typo for ‘rho’ was corrected and the caption of table 2 was updated as:
“Raman mode assignments of vibrational modes in 10-3500 cm-1 frequency range according to PbCl6 octahedra and MA cation molecular symmetry. All the vibrational modes correspond to the orthorhombic phase .”
- There are three peaks 68, 2920 and 3201 cm-1 were not assigned or found in the previous works.
- The three peaks at 68, 2920 and 3201 cm-1 were not found in previous works, hence not assigned to any vibrational mode. The information related to this was added at page 8 (first paragraph, last line) as:
“Three peaks located at 68, 2920 and 3201 cm-1 could not be found in any of previous reports, hence were not assigned to any vibrational mode.”
- Paragraph 6, section 3.1 mentioned two different Orthorhombic phases were identified in the previous works. What's your Orthorhombic phase?
- Previously the orthorhombic phase for MAPbCl3 was designated to space group. However, Chi al. proved it to be [1], which is widely adapted in the current research era. We also adapt the proposition of Chi et.al. In the manuscript following statement at page 8 (last paragraph, fourth line) refers to the above story:
“Regarding the orthorhombic phase, was suggested earlier [24], but, more recently, was proposed [32].”
- The region around tetragonal phase in Figure 6, 7, &8 should be zoomed in for clearer view. The same things for Figure S 1, 2, &3.
- Firstly, for figure 6 where we drew four subfigures already, we can note that all transitions are clearly visible in the current view. However, in figure 7, for a better and clear view, it was updated from two subfigures (a and b) to three subfigures (a,b and c). The caption of figure 7 was also updated accordingly as stated below:
“Temperature dependences of the Raman shift in a 400-1700 cm-1 frequency range (a-c). Phase boundaries are shown as vertical dotted lines along with the phase transition temperatures.”
For the figure 8 and the supplementary figures S1, 2 and 3, we think the subfigures are already evident enough to clearly see anomalous behavior in the Raman shift and FWHMs near the phase transition temperatures.
- Figure 6 does not show mode symbols.
- The respective mode symbols were added for figure 6.
- Caption of Figure 9 does not mention the temperature to measure the Raman spectral for PbCl2 and MACl.
- The temperature at which Raman spectra were measured is already mentioned in the first line of caption for figure 9 represented below as bold letters:
“Raman spectra of degraded MAPbCl3 compared with that of its by-products at RT.”
However, for more clarity RT has been changed to room temperature, and the names of by-products have been added in the caption as shown below:
“Raman spectra of degraded MAPbCl3 compared with that of its by-products (PbCl2 and MACl) at room temperature.”
- The discussion and comparison with theoretical works (if available) might be helpful to determine the phonon modes. For instance, there have been two detailed calculations for MAPbI3 [PHYSICAL REVIEW B 92, 144308 (2015)], [J. Phys. Chem. C 122, 21703 (2018)]
- Thank you for your comment. For the known articles on theoretical calculations on MAPbCl3, results were compared at certain points in the manuscript as shown below:
Page 13 (second paragraph, tenth line)
“Interestingly, such mode splitting is more common in Cl-based halide perovskites according to theoretical analyses [20,37].”
Page 14 (first paragraph, fourth line)
“A similar mode was observed by Nguyen et.al (at 2957 cm-1 in the experiment and 2974 cm-1 via theoretical calculations) which they associated with the MA vibration in MACl [17].”
Reference
[1] Chi, L.; Swainson, I.; Cranswick, L.; Her, J.-H.; Stephens, P.; Knop, O.; Chi, L.; Swainson, I.; Cranswick, L.; Her, J.-H.; et al. The Ordered Phase of Methylammonium Lead Chloride CH3ND3PbCl3. J. Solid State Chem. 2005, 178, 1376–1385.
We hope the present version is satisfactory for its publication in Materials.
Your sincerely,
Jae-Hyeon Ko.

Round 2
Reviewer 1 Report
The current version is very good and can be published as it is.
Reviewer 2 Report
I fairly agree with all the changes in the manuscripts.